# Impact of Industrial Structure Upgrading on Green Total Factor Productivity in the Yangtze River Economic Belt

**DOI:** 10.3390/ijerph19063718

**Published:** 2022-03-21

**Authors:** Jinhua Sun, Decai Tang, Haojia Kong, Valentina Boamah

**Affiliations:** 1School of Law and Business, Sanjiang University, Nanjing 210012, China; sunjh@sju.edu.cn; 2School of Management Science and Engineering, Nanjing University of Information Science & Technology, Nanjing 210044, China; 20215242005@nuist.edu.cn; 3School of Economics and Management, Nanjing University of Science & Technology, Nanjing 210000, China

**Keywords:** green total factor productivity, industrial structure, environmental regulation, Yangtze River economic belt

## Abstract

The Yangtze River economic belt is an inland river economic belt with international influence composed of 11 provinces and municipalities in the Yangtze River Basin. This paper uses the super-efficiency model to calculate the green total factor productivity of 11 provinces and municipalities in the Yangtze River economic belt (YREB). Then we establish a model to study the impact of industrial structure upgrading, industrial structure rationalization, and environmental regulation on green total factor productivity (GTFP). Empirical analysis shows that the industrial structure upgrading and environmental regulation have a significant impact on GTFP and show regional characteristics. The more developed the economy and the higher the industrial structure, the greater the impact of upgrading and environmental regulation on GTFP. Compared with other control variables, the urbanization rate impacts GTFP, followed by regional economic development.

## 1. Introduction

Promoting the development and construction of the Yangtze River economic belt (YREB) is an important national strategy of China. That is, it prioritizes regional green development and ecological green development at home and abroad, promoting coordinated, balanced, and innovation-driven high-quality development [1]. The YREB spans Eastern, Central, and Western China, including 11 provinces (municipalities) such as Shanghai, Jiangsu, Zhejiang, Anhui, Jiangxi, Hubei, Hunan, Chongqing, Sichuan, Yunnan, and Guizhou (as shown in Figure 1). It is an important support for China’s economy.

In 2020, the regional GDP exceeded CNY 47 trillion, accounting for 46.5% of the country. Figure 2 shows the proportion and change trend of three industries in the YREB from 2005 to 2020. In 2020, the three industries in the YREB accounted for 7.23%, 38.74%, and 54.02% of the regional GDP, respectively. The industrial structure shows a trend of continuous upgrading.

The YREB has rich natural resources, a strong industrial foundation, significant talent advantages, and strong economic growth potential. However, for a long time, the proportion of iron and steel, petrochemical, cement, and other industries in the industrial structure has been high, and there is great pressure to upgrade the industrial structure. To effectively improve the quality of economic growth, the National Development and Reform Commission, the Ministry of Science and Technology, and the Ministry of Industry and Information Technology of China jointly issued the plan for the YREB’s industrial transformation and upgrading driven by innovation in September 2016. The document proposed that “promoting industrial transformation and upgrading driven by innovation is an important task for the YREB to achieve economic quality, efficiency and green development”. With the slowdown of economic growth, as an important economic belt in China, the sustainable economic growth of the YREB in the future should improve the quality of economic growth [2]. The economic development of 11 provinces (municipalities) in the YREB is very unbalanced. Promoting the green and sustainable development of the YREB is an important issue [3].

This paper aims to study the impact of industrial structure upgrading of the YREB on green total factor productivity (GTFP). We collected the relevant data of 11 provinces (municipalities) in the YREB from 2005 to 2019 for analysis. Firstly, considering the undesirable output such as environmental pollution, we use the super-efficiency SBM model to calculate the value of GTFP in 11 provinces (municipalities). We find that GTFP presents regional characteristics in the YREB. Then, we establish a model to study the impact of the upgrading and rationalization of industrial structure on GTFP. The results show that industrial structure upgrading has a significant positive impact on GTFP, while industrial structure rationalization has no significant impact on GTFP. At the same time, environmental regulation, as a regulatory variable, also has a significant impact on GTFP. After further analysis, it is found that the impact of industrial structure upgrading and environmental regulation on GTFP in the eastern region is significantly higher than that in the western region. At the same time, we also observed that urbanization also promoted the improvement of GTFP. Finally, we review and prospect the analysis. The YREB should promote upgrading industrial structures, protect the environment, and promote urbanization to improve GTFP.

This paper makes contributions to the literature from the following aspects. Firstly, our paper provides new evidence for understanding the relationship between upgrading industrial structure and GTFP. Promoting the upgrading of industrial structure can promote energy conservation and environmental protection to improve GTFP. Secondly, this paper provides suggestions for the development of the YREB. We believe that the economic development of provinces (municipalities) in the YREB is unbalanced, and the industrial structure has different effects on GTFP in different regions. Therefore, provinces and municipalities should adjust measures to local conditions to promote industrial structure upgrading and environmental protection [4,5].

## 2. Review of Literature

Green total factor productivity (GTFP), which focuses on green growth, includes traditional labor, capital, and economic input and considers energy consumption and environmental pollution [6]. The upgrading of the industrial structure means that in economic growth, the proportion of output value and employment of the primary industry decreases [7]. In contrast, the proportion of output value and employment of the secondary and tertiary industries increases, and there is a trend of unbalanced growth among various industries. For a long time, scholars have believed that the flow of production factors in various industrial sectors promotes the industrial structure upgrading and improves the efficiency of production factors and economic growth.

### 2.1. GTFP Measurement

Many scholars calculated GFTP through data envelopment analysis (DEA). Kong [8] studied the total factor productivity of ten major industries in Singapore using the nonparametric frontier method of DEA and obtained the Malmquist productivity index at the sectoral level. Kuosmanen et al. [9] analyzed the panel data of 459 Finnish farms from 1992 to 2000 and revealed the driving factors behind productivity changes. Hu [10] used the DEA Malmquist index model to measure the green Malmquist index of 29 provinces in China from 1995 to 2008. Wang [11] selected DEA-BCC and DEA-Malmquist models to conduct empirical measurement and comparative research on temporal and spatial differentiation of green economy efficiency of 285 cities in China from 2004 to 2012. Feng [12] used the SBM-DEA method to study and found that China’s GTFP showed a decreasing trend from the eastern coastal region to the western region. Wang [13] used SBM directional distance function and the global Malmquist–Luenberger index to estimate the change and decomposition of the GTFP of the service industry in various regions of China from 2002 to 2014. Ren [14] used the DEA Malmquist index method to measure China’s provinces’ GTFP and decomposition index from 2007 to 2016 and introduced a spatial Durbin econometric model to analyze the spatial spillover effect and regional influencing factors GTFP. Chen [15] applied a three-stage data envelopment analysis method combined with the slack-based measure model to eliminate the influences of environmental factors and random errors and explore the real agricultural GTFP of 30 provinces in China from 2000 to 2017. Li [16] proposed slack based measure-Malmquist–Luenberger (SBM-ML) model to measure GTFP of the pearl river delta urban agglomeration from 2005 to 2018. Xiao [17] estimated the GTFP of 33 industrial sectors in China from 2002 to 2014 based on the nonparametric DEA-Malmquist method. DAR [18] used nonparametric data envelopment analysis (DEA) technology to analyze the technical efficiency and scale efficiency scores of Indian banks. Zhang [19] calculated one belt, one road, total factor productivity of 36 countries by using the DEA-Malmquist index method and established a dynamic panel model.

### 2.2. Industrial Structure and GTFP

William Petty [20] was the first to notice the evolution of industrial structure. According to Petty, he believes that “the income of industry is more than that of agriculture, and the income of commerce is more than that of industry”. This income gap between industries will promote the transfer of the labor force from low-income industries to high-income industries and promote economic growth at the same time. Combes [7] analyzed the data of manufacturing and service industries in France. He concluded that different industrial structures have different effects on economic growth. Buera and Kaboski [21] used a standard neoclassical growth model to analyze the agriculture, manufacturing, and service industries in the United States from 1870 to 2000. They believed that the professional and highly skilled labor force had made a significant contribution to economic growth. Noseleit [22] used the data of Germany from 1975 to 2002, and he considered that inter-industry redistribution is an important means to accelerate economic growth. Jebali [23] investigated the environmental productivity of Mediterranean countries from 2009 to 2014. He considered that technological progress and industrial structure upgrading are the main sources of productivity growth. Fukao [24] studied Japan’s structural transformation and found that the commerce and service industry played a driving role in Japan’s GDP growth throughout the 20th century. Li [25], Liu [26], Zhang [27], She [28], and Zhang [29] found that the upgrading of the industrial structure plays a positive role in promoting GTFP through the empirical study. Strengthening the upgrading of industrial structure and effective allocation of resources can promote the improvement of GTFP [30].

Scholars have studied the impact of industrial structure adjustment on economic development through different research methods in the existing research. Most scholars affirm that industrial structure upgrading plays a role in promoting economic growth and explore scientific methods to study the impact of industrial structure upgrading of the YREB across different stages of economic development on GTFP. It can provide suggestions for adjusting the regional industrial structure, promoting optimal allocation of resources among regions, improving regional innovation and development, and supporting green all-around development.

## 3. Measurement of GTFP

In this section, GTFP will be calculated first.

### 3.1. Measurement Method

Total factor productivity refers to the utilization efficiency of all production factors input in a certain period [6]. Taking natural resources and ecological environment factors into account, the GTFP is formed. GTFP should consider not only desirable output but also undesirable output such as environmental pollution. Therefore, GTFP is an important performance index to measure regional sustainable development. Data envelopment analysis (DEA) is a commonly used method to calculate GTFP because it can calculate the decision-making units (DMU) with more input and output. In this paper, a non-radial super-efficiency SBM model [31,32], including undesirable output, is established to measure GTFP. Assuming that each province is a decision-making unit (DMU), there are *n* DMUs in total. The input x, desirable output $y^{g}$, and undesirable output $y^{b}$ of each DMU are as follows:$$x\in R^{m}$$
$$y^{g}\in R^{s1}$$
$$y^{b}\in R^{s2}$$

Then, there are matrixes:$$X=\left[x_{1},x_{2},\ldots ,x_{n}\right]\in R^{m\times n}$$
$$Y^{g}=\left[y_{1}^{g},y_{2}^{g},\ldots ,y_{n}^{g}\right]\in R^{s1\times n}$$
$$Y^{b}=\left[y_{1}^{b},y_{2}^{b},\ldots ,y_{n}^{b}\right]\in R^{s2\times n}$$

SBM model formula is expressed as
(1)$$\rho =min\frac{1-\frac{1}{m}\sum_{i=1}^{m}\frac{s_{i}^{-}}{x_{ik}}}{1+\left[\frac{1}{s_{1}+s_{2}}\left(\sum_{r=1}^{s_{1}}\frac{s_{r}^{g}}{y_{rk}^{g}}+\sum_{r=1}^{s_{2}}\frac{s_{r}^{b}}{y_{rk}^{b}}\right)\right]}$$

Subject to
$$x_{k}=X\lambda +s^{-}$$
$$y_{k}^{g}=Y^{g}\lambda -s^{g}$$
$$y_{k}^{b}=Y^{b}\lambda +s^{b}$$
$$\lambda \geq 0,s^{-}\geq 0,s^{g}\geq 0,s^{b}\geq 0$$
where *m* is the number of input indicators, $s_{1}$ and $s_{2}$ are the numbers of desirable output indicators and undesirable output indicators, respectively, and k is the number of time. $s^{-},\;s^{g},\;s^{b}$ are relaxation variables of input, desirable output, and undesirable output, respectively. λ is the coefficient vector of the DMU linear combination. *ρ* is the efficiency value of DMU. If ρ=1 that indicates that the DMU is efficient, and ρ<1 indicates that the DMU is inefficient. Suppose ρ=1 super-efficiency analysis is carried out to calculate the specific value.

$Px_{0},y_{0}$ is an efficient production possibility set.
$$Px_{0},y_{0}=\left\{\left(\bar{x},\bar{y}\right)|\bar{x}\geq \sum_{j=1,\neq 0}^{n}\lambda_{j}x_{j},\bar{y}\leq \sum_{j=1,\neq 0}^{n}\lambda_{j}y_{,}\bar{y}\geq 0,\lambda \geq 0\right\}$$

The super efficiency SBM formula is as follows:(2)$$\rho^{*}=min\frac{\frac{1}{m}\sum_{i=1}^{m}\frac{{\bar{x}}_{i}}{x_{i0}}}{\frac{1}{s_{1}+s_{2}}\left(\sum_{r=1}^{s_{1}}\frac{{\bar{y}}_{r}^{g}}{y_{r0}^{g}}+\sum_{r=1}^{s_{2}}\frac{{\bar{y}}_{r}^{b}}{y_{r0}^{b}}\right)}$$

Subject to
$$\bar{x}\geq \sum_{j=1,\neq 0}^{n}\lambda_{j}x_{j},$$
$${\bar{y}}^{g}\leq \sum_{j=1,\neq 0}^{n}\lambda_{j}y_{j}^{g},$$
$${\bar{y}}^{b}\geq \sum_{j=1,\neq 0}^{n}\lambda_{j}y_{j}^{b},$$
$$\bar{x}\geq x_{0},{\bar{y}}^{g}\leq y_{0}^{g},{\bar{y}}^{b}\leq y_{0}^{b},\lambda \geq 0$$

The value of ρ and $\rho^{*}$ can be used to represent GTFP.

### 3.2. Selection of Measurement Variables

According to the super-efficiency SBM model, labor input, capital input, and energy consumption are selected as input indicators based on the scientificity and availability of data. Regional GDP and urban green space area are selected as desirable output indicators, and wastewater emission and industrial sulfur dioxide emission are selected as undesirable output indicators. This paper selects the relevant data from 2008 to 2019 for calculation. The variables are listed in Table 1.

### 3.3. Measurement Result

According to the aforementioned SBM model, the above variables are used to calculate the GTFP of provinces and municipalities in the YREB. The results are shown in Table 2.

It can be seen from Table 2 that the GTFP of provinces in the YREB is the lowest in the west and the highest in the east. Among them, the GTFP of Shanghai and Jiangsu Province is always higher than 1, indicating that the investment of these two provinces and municipalities is the most efficient. The GTFP of Anhui Province, Hubei Province, Guizhou Province, and Yunnan Province shows a downward trend yearly. From the perspective of efficiency decomposition, it is mainly due to the decline of technological progress efficiency [35].

## 4. Empirical Research Design

### 4.1. Model Construction

Based on theoretical analysis, to explore the impact of industrial structure upgrading on GTFP, an econometric model is adopted as follows:(3)$$lnGTFP_{it}=\alpha_{0}+\beta_{1}lnISU+\beta_{2}lnISR+\beta_{i}lnCTRL_{it}+\varepsilon_{it}$$
where *i* and *t* indicate the province and period, $\alpha_{0}$ indicates the section effect, $\beta_{i}$ represents the regression coefficient of each variable, and ε is a random error term.

The regulatory variable environmental regulation is introduced to build the model as follows:(4)$$lnGTFP_{it}=\lambda_{0}+\lambda_{1}lnISU+\lambda_{2}lnISR+\lambda_{3}lnER+\lambda_{2}lnCTRL_{it}+\varepsilon_{it}$$
where *i* and *t* indicate the province and period, $\lambda_{0}$ indicates the section effect, $\lambda_{i}$ represents the regression coefficient of each variable, and ε is a random error term.

### 4.2. Variable Selection

#### 4.2.1. Explained Variable: GTFP

The GTFP of each province (municipality) is calculated according to the above method. The value of GTFP is shown in Table 2.

#### 4.2.2. Explanatory Variables: Industrial Structure

The industrial structure is the composition of various industries in the national economy. The industrial structure can be considered from two aspects: the industrial structure upgrade and the industrial structure rationalization.

The industrial structure upgrade is the upgrading process of industries and the embodiment of the proportional relationship between industries. The upgrading index is determined according to the following formula:(5)$$ISU=\sum V_{it}^{n}\times LP_{it}^{N}$$
where $V_{it}^{n}$ is the proportion of the output value of industry *i* in the regional GDP in the *t* year of region *n* and $LP_{it}^{N}$ represents the standardized labor productivity of industry *i* in year *t* of region *n*. The calculation formula of $LP_{it}^{N}$ is as follows:(6)$$LP_{it}^{N}=\frac{LP_{it}^{n}-LP_{ib}}{LP_{if}-LP_{ib}}$$
where $LP_{it}^{n}$ represents the labour productivity of industry *i* in year *t* of region *n*, $LP_{ib}$ represents the labour productivity of industry *i* at the beginning of industrialization, and $LP_{if}$ represents the labor productivity of industry *i* at the completion of industrialization. Referring to Chenery [36] and Liu [37], the labor productivity at the beginning of industrialization is CNY 2570 for the primary industry, CNY 10,755 for the secondary industry, and CNY 12,509 for the tertiary industry. The labor productivity in the completion of industrialization is CNY 53,058 for the primary industry, CNY 141,036 for the secondary industry, and CNY 49,441 for the tertiary industry at the end of industrialization (all calculated at 2005 prices).

Industrial structure rationalization refers to the coupling degree of each industry’s factor input structure and output structure. It is measured by the Theil index. The formula is as follows:(7)ISR=∑i=1nYiYLNYiLiYL
where $Y_{i}$ represents the output value of the industry *i*, *Y* represents the total output value of the region, $L_{i}$ is the number of employees in the industry *i*, *L* is the total number of employees in the province, and n represents the number of industrial sectors. When ISR = 0, the regional industrial structure is reasonable. The larger the index, the worse the degree of industrial structure rationalization.

#### 4.2.3. Regulatory Variable: Environmental Regulation

Environmental regulation is a kind of social regulation [38]. It is a legal policy, and the government adopts its implementation process to restrict economic activities in order to protect the environment [39]. Environmental regulation is measured by the comprehensive index of the discharge of industrial pollutants (industrial wastewater, industrial sulfur dioxide, and industrial solid waste) and the total industrial output value. The formula is as follows [40,41,42]:(8)ERi,t=113∑n=13En,it=113∑n=13en,itYiten,tYt
where $E_{n,it}$ refers to the relative position of the *n*-th pollutant emission intensity in the whole county of province *i* in *t* year, $e_{n,it}$ represents the emission of the nth pollutant of province *i* in *t* year, $Y_{it}$ refers to the total industrial output value of province *i* in *t* year, $e_{n,t}$ represents the national emission of the *n*-th pollutant in *t* year, and $Y_{t}$ represents the total industrial output value of the country in *t* year.

The higher the ER value, the stricter the environmental standards implemented by the government and the stronger the environmental regulation.

#### 4.2.4. Control Variables

The control variables selected in this study include (1) economic development level, which is measured by per capita regional GDP, (2) the degree of opening to the outside world, which is measured by the ratio of export trade volume to regional GDP, (3) local government input, which is measured by the ratio of local government financial expenditure to regional GDP, and (4) urbanization rate, which is measured by the proportion of the urban population in the total population.

The variables are listed in Table 3.

### 4.3. Data Sources and Descriptive Statistics

Taking the panel data of 11 provinces and municipalities in the YREB from 2005 to 2019 as the sample, the data are derived from the statistical yearbook of each province and municipality and the statistical yearbook of China.

The descriptive statistics are shown in Table 4.

## 5. Empirical Research Results and Analysis

### 5.1. Main Regression Results

Table 5 shows the main regression results of the impact of industrial structure upgrading on GTFP in 11 provinces and municipalities of the YREB.

From the specific direction and regression results, the explanatory variables, regulatory variables, and control variables show the same direction and significance. The explanatory variable industrial structure upgrading (ISU) is significantly positive at the 1% confidence level. This means that the upgrading of the industrial structure plays a significant role in promoting GTFP [43]. The industrial structure upgrade is the increase in the proportion of secondary industry and tertiary industry. The development of the tertiary industry has driven the increase of more pollution-free enterprises, promoted the reallocation of resources, and reduced the emission of pollution to improve the GTFP. Moreover, after adding the environmental regulation (ER) variable, the coefficient of ISU has increased. This shows that, with the enhancement of environmental awareness and the improvement of environmental protection regulations in various regions, enterprises in various regions have strengthened self-discipline and minimized the emission of pollutants. This indicates that the industrial structure upgrading has a significant spillover effect on GTFP [44]. The explanatory variable industrial structure rationalization (ISR) estimation result is not significant. The industrial structure rationalization measures each industry’s coupling degree of output and input and has no significant promotion or inhibition effect on the GTFP. The regulatory variable environmental regulation (ER) is significantly positive at the 1% confidence level, which shows that with the attention paid to environmental protection in various regions, enterprises pay attention to technology research and development in industry upgrading, update backward technologies and equipment, reduce pollutant emissions, facilitate the growth of economic quality, and promote GTFP. Among the control variables, economic development level (EDL), local government input (INP), and urbanization rate (UR) are significantly correlated with GTFP at the confidence level of 1%.

### 5.2. Subregional Test

Per the 11 provinces and municipalities of the YREB span from the west to the east of China, the economic development of each region is uneven. The YREB is divided into three regions: west, midland, and east. The results are shown in Table 6.

The results of Table 6 show that the industrial structure has different effects on GTFP in the west, middle, and east of the YREB. Still, ISU has a significant positive impact on GTFP [17], and the coefficient from west to east shows an increasing trend. While ISR does not pass the regression test in the west, it has a significance of 5% and 1% in the middle and east. It indicates that the rationality of the industrial structure in the western region is insufficient. ER also has a significant impact on GTFP, and the higher the degree of economic development, the greater the impact of ER. The urbanization rate in the control variables passed the significance check. Still, the coefficient values in different regions are different, indicating that the role of the urbanization process in GTFP is quite different in different regions.

## 6. Research Conclusions and Suggestions

### 6.1. Research Conclusions

Through the study on the GTFP of 11 provinces and municipalities in the YREB from 2005 to 2019, it is found that the GTFP value of Shanghai is the highest, while that of Yunnan is the lowest. The efficiency value of GTFP of provinces and municipalities in the YREB in the east is higher than that in the middle and west (as shown in Figure 3), which is consistent with the level of regional economic development.

The industrial structure upgrading index calculated by formulas (5) and (6) is shown in Figure 4. Shanghai, Jiangsu, and Zhejiang indexes in the east of the YREB are the highest, while those of Yunnan, Sichuan, Guizhou, and Jiangxi in the west are lower. Chongqing in the west ranks fourth after provinces and municipalities in the east, reflecting its role as an economic center and modern manufacturing base in the upper reaches of the Yangtze River.

The empirical analysis of the industrial structure upgrade on GTFP in 11 provinces and municipalities of the YREB from 2005 to 2019 found that the industrial structure upgrade plays a significant role in promoting GTFP [45]. The impact of industrial structure upgrading on GTFP is deepened under environmental regulation. From different regions, compared with underdeveloped regions, the industrial structure upgrade is more conducive to the improvement of GTFP in economically developed regions. Therefore, promoting the advanced development of industrial structures and formulating environmental protection laws and regulations in line with regional development are important ways to promote sustainable economic development and improve GTFP.

### 6.2. Countermeasures and Suggestions

#### 6.2.1. Promote the Industrial Structure Upgrade in Various Regions of the YREB

The industrial structure upgrade has a significant impact on GTFP. As such, all regions should formulate appropriate industrial policies in combination with the regional resource situation and economic development stage [46]. As a whole, the YREB should pay attention to the complementarity of industrial layout and regional economy according to the carrying capacity of regional resources and environment. The industrial structure of eastern provinces and municipalities should develop to high-tech and intelligent manufacturing industry to explore the demonstration role. The central and western provinces should accelerate the upgrading and transformation of traditional industries, improve the green and intelligent level of industries, promote the development of producer services, extend the manufacturing service chain, and promote the upgrading of the industrial structure of the YREB.

#### 6.2.2. Promote Environmental Protection

The government should formulate perfect environmental protection laws and regulations and strengthen pollution control [46]. Many petrochemical, iron and steel, and non-ferrous metal enterprises in the provinces and municipalities of the YREB have accelerated the technological upgrading and equipment transformation of these enterprises in multiple ways. This has also promoted the application of new technologies, new equipment, and new materials for energy-saving, water-saving, and cleaner production [35]. They should vigorously develop the energy conservation and environmental protection industry, improve the technology R&D and services of energy conservation and environmental protection industry, and promote the centralized development of energy conservation and environmental protection equipment manufacturing industry.

#### 6.2.3. Push Forward the New Urbanization of the YREB

The scale and agglomeration effect of cities have an important impact on GTFP [47]. All provinces and municipalities should make and improve attractive talent introduction policies according to the economic level and development orientation, appropriately control the expansion speed of old urbanization, improve the quality of urbanization, and promote the balanced development of human capital and GTFP [48].

### 6.3. Limitations and Future Research

Although this paper provides sufficient evidence for the impact of industrial structure upgrading of the YREB on GTFP, it also has limitations for further research. Firstly, this study uses the data envelopment analysis method to measure the GTFP. The result is a static efficiency value, so the result is not comprehensive enough, and it will be improved in future research. Secondly, this study divides industrial structure upgrading into industrial structure upgrading and industrial structure rationalization. In further research, the calculation of industrial structure upgrading can be further expanded. Finally, this paper takes 11 provinces (municipalities) in the YREB as the research object and draws some conclusions. Among these 11 provinces (municipalities), there is a large gap among the constituent cities. Further exploration and research should be made on the adjustment of industrial structure.

## Figures and Tables

**Figure 1 ijerph-19-03718-f001:**
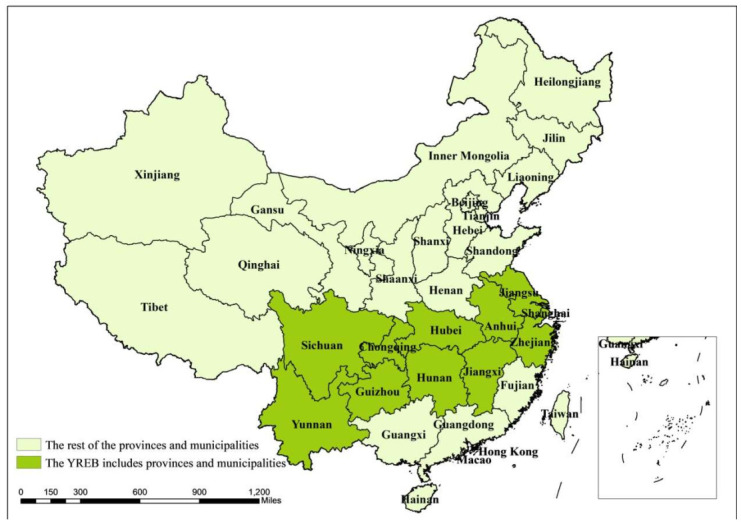
The Yangtze River economic belt research area (dark green).

**Figure 2 ijerph-19-03718-f002:**
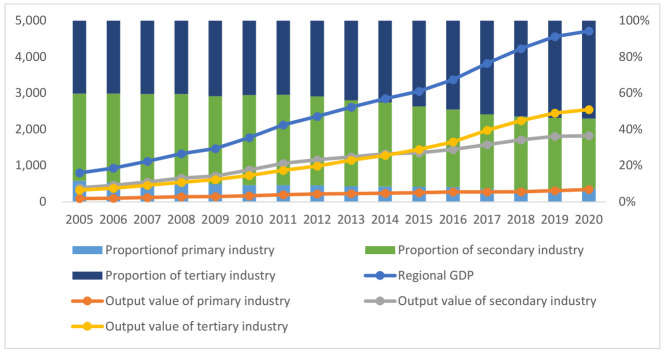
Three industries’ output value (CNY billion) and structure (%) of YREB (data source: Statistical Yearbooks of provinces and municipalities).

**Figure 3 ijerph-19-03718-f003:**
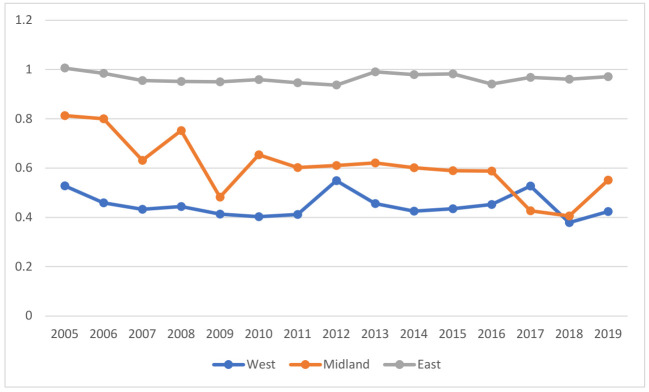
GTFP value.

**Figure 4 ijerph-19-03718-f004:**
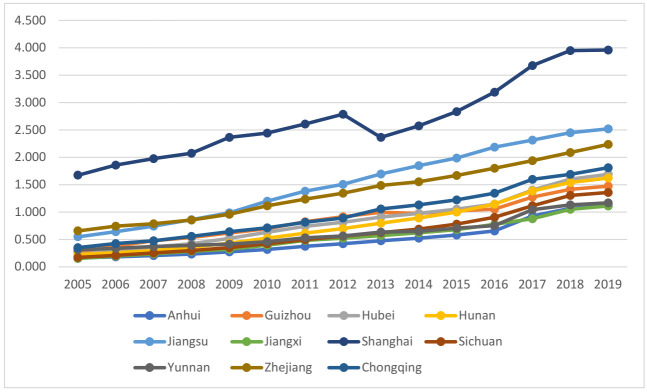
Industrial structure upgrading index.

**Table 1 ijerph-19-03718-t001:** GTFP measurement variables.

Primary Index	Secondary Index	Variable Description
Input	Labor input	Number of employed persons in the region (10^4^ persons)
	Capital input	Calculate the capital stock by using the perpetual inventory method (CNY 10^8^) ^1^
	Energy consumption	Electricity consumption (100 million kwh)
Desirable output	Regional GDP	Regional GDP (CNY 10^8^)
	The urban green space area	Urban green space area (10^4^ hectares)
Undesirable output index	Wastewater emission	Total industrial wastewater discharge (100 million tons)
	Industrial sulfur dioxide emission	Industrial sulfur dioxide emission (100 million tons)

^1^ The capital input adopts the perpetual inventory method to calculate the capital stock, and the formula is $K_{i,t}=K_{i,t-1}\left(1-\delta_{i,t}\right)+I_{i,t}$, where $K_{i,t}$ represents the capital stock of province *i* in year *t*, $\delta_{i,t}\;$ represents the depreciation rate of province *i* in year *t*, and $I_{i,t}$ refers to the total investment in fixed assets of province *i* in year *t*. The capital stock in the base period is 2004, and the calculation formula is $K_{i,2004}=\frac{I_{i,2005}}{g_{i,\left(2005-2019\right)}+\delta }$. The fixed asset investment amount in each year is reduced to the constant price in 2005 by using the fixed asset investment price index of each province, and δ  is supposed to 9.6% (Zhang [33,34]).

**Table 2 ijerph-19-03718-t002:** GTFP in all provinces/municipalities of the YREB.

**Year**	**West**				
	**Yunan**	**Guizhou**	**Sichuan**	**Chongqing**	**Mean**
2005	0.546	0.436	0.544	0.585	0.528
2006	0.45	0.416	0.498	0.473	0.459
2007	0.411	0.411	0.456	0.454	0.433
2008	0.405	0.427	0.469	0.476	0.444
2009	0.367	0.393	0.431	0.462	0.413
2010	0.347	0.371	0.431	0.464	0.403
2011	0.33	0.357	0.452	0.508	0.412
2012	0.338	0.367	0.482	1.01	0.549
2013	0.35	0.406	0.506	0.563	0.456
2014	0.321	0.373	0.472	0.534	0.425
2015	0.315	0.389	0.46	0.575	0.435
2016	0.295	0.395	0.451	0.665	0.452
2017	0.328	0.348	0.422	1.008	0.527
2018	0.316	0.332	0.411	0.458	0.379
2019	0.329	0.344	0.541	0.483	0.424
mean	0.363	0.384	0.468	0.581	0.449
**Year**	**Midland**				
	**Hubei**	**Hunan**	**Anhui**	**Jiangxi**	**Mean**
2005	0.599	1.027	0.61	1.015	0.813
2006	0.57	1.032	0.589	1.01	0.800
2007	0.512	1.033	0.47	0.508	0.631
2008	0.523	1.03	0.453	1.003	0.752
2009	0.513	0.536	0.421	0.456	0.482
2010	0.519	1.001	0.424	0.672	0.654
2011	0.517	1.009	0.431	0.451	0.602
2012	0.537	1.017	0.431	0.454	0.610
2013	0.556	1.028	0.433	0.468	0.621
2014	0.536	1.025	0.406	0.436	0.601
2015	0.527	1.032	0.385	0.41	0.589
2016	0.558	1.027	0.379	0.386	0.588
2017	0.484	0.508	0.365	0.35	0.427
2018	0.463	0.469	0.354	0.339	0.406
2019	0.458	1.038	0.353	0.354	0.551
mean	0.525	0.921	0.434	0.554	0.608
**Year**	**East**				
	**Jiangsu**	**Shanghai**	**Zhejiang**	**Mean**	
2005	1.043	1.223	0.751	1.006	
2006	1.032	1.237	0.683	0.984	
2007	1.011	1.241	0.613	0.955	
2008	1.001	1.241	0.611	0.951	
2009	1.019	1.254	0.577	0.950	
2010	1.018	1.262	0.598	0.959	
2011	1.023	1.217	0.597	0.946	
2012	1.029	1.185	0.596	0.937	
2013	1.044	1.192	0.733	0.990	
2014	1.045	1.206	0.685	0.979	
2015	1.051	1.268	0.628	0.982	
2016	1.057	1.211	0.556	0.941	
2017	1.039	1.413	0.453	0.968	
2018	1.029	1.417	0.435	0.960	
2019	1.045	1.412	0.456	0.971	
mean	1.032	1.265	0.598	0.965	

**Table 3 ijerph-19-03718-t003:** Variable description.

Variable Type	Name	Code	Description
Explained variable	Green total factor productivity	GTFP	According to 3.3 measurement result
Explanatory variables	Industry structure upgrade	ISU	Calculated according to formula (5)
	Industry structure rationalization	ISR	Calculated according to formula (7)
Regulatory variable	Environmental regulation	ER	Calculated according to formula (8)
Control variable	Economic development level	EDL	EDL = Per capita GDP (CNY 10^4^)
	Degree of openness	EXP	ESP = Regional export trade volume/regional GDP
	Local government input	INP	INP = Regional government expenditure/regional GDP
	Urbanization rate	UR	UR = Regional urban population/total population

**Table 4 ijerph-19-03718-t004:** Descriptive statistics.

Variable	Obs	Mean	Std. Dev.	Min	Max
GTFP	165	0.624	0.295	0.206	1.417
ISU	165	0.978	0.633	0.154	3.142
ISR	165	0.207	0.178	0.001	0.819
ER	165	1.331	0.859	0.352	5.772
EDL	165	4.290	2.999	0.505	15.659
EXP	165	0.187	0.206	0.020	0.899
INP	165	0.207	0.071	0.090	0.402
UR	165	0.524	0.148	0.269	0.893

**Table 5 ijerph-19-03718-t005:** Regression results of industrial structure affecting GTFP.

Variables	Model 1	Model 2	Model 3	Model 4
Ln ISU	0.154 ***	0.203 *	0.282 ***	0.328 ***
	(4.773)	(1.935)	(7.661)	(3.039)
Ln ISR	−0.047	−0.002	−0.030	−0.005
	(−1.135)	(−0.059)	(−0.806)	(−0.119)
Ln ER			0.511 ***	0.319 ***
			(6.018)	(3.275)
Ln EDL		0.687 ***		0.571 ***
		(4.136)		(3.434)
Ln EXP		0.117 ***		0.054
		(2.732)		(1.181)
Ln INP		−0.528 ***		−0.405 ***
		(−4.390)		(−3.280)
Ln UR		−1.406 ***		−0.998 ***
		(−4.188)		(−2.792)
Constant	−0.667 ***	−2.778 ***	−0.725 ***	−2.406 ***
	(−5.165)	(−6.376)	(−6.627)	(−5.376)
Observations	165	165	165	165
Number of DMU	11	11	11	11

Z-statistics in parentheses: *** *p* < 0.01, ** *p* < 0.05, * *p* < 0.1.

**Table 6 ijerph-19-03718-t006:** Regression results of the impact of subregional industrial structure on GTFP.

VARIABLES	West	Midland	East
Ln ISU	0.309 ***	0.690 **	2.327 ***
	(3.083)	(1.943)	(3.533)
Ln ISR	−0.137	0.065 **	0.772 ***
	(−1.065)	(2.503)	(4.860)
Ln ER	0.412 ***	0.039 *	0.691 ***
	(3.208)	(0.212)	(3.219)
Ln EDL	−0.723 ***	0.072 *	−3.273 ***
	(−2.868)	(1.034)	(−3.885)
Ln EXP	0.068	−0.268 *	−0.409
	(1.062)	(−1.878)	(−1.507)
Ln INP	−0.126	−0.134	−0.528
	(−0.669)	(−0.446)	(−1.300)
Ln UR	0.478	−3.975 ***	2.272 ***
	(1.215)	(−3.362)	(2.915)
Constant	0.076	−3.907 ***	5.405 ***
	(0.137)	(−3.393)	(3.631)
Observations	60	60	45
Number of DMU	4	4	3

Z-statistics in parentheses: *** *p* < 0.01, ** *p* < 0.05, * *p* < 0.1.

## Data Availability

All the original data are from the official websites of China Statistics Bureau (http://www.stats.gov.cn/tjsj/ndsj/, accessed on 15 February 2022); Anhui Statistics Bureau (http://tjj.ah.gov.cn/ssah/qwfbjd/tjnj/index.html, accessed on 15 February 2022); Guizhou Statistics Bureau (http://stjj.guizhou.gov.cn/, accessed on 15 February 2022); Hubei Statistics Bureau (http://tjj.hubei.gov.cn/tjsj/sjkscx/tjnj/qstjnj/, accessed on 15 February 2022); Hunan Statistics Bureau(http://tjj.hunan.gov.cn/tjsj/tjnj/, accessed on 15 February 2022); Jiangsu Statistics Bureau (http://tj.jiangsu.gov.cn/col/col83749/index.html, accessed on 15 February 2022); Jiangxi Statistics Bureau (http://tjj.jiangxi.gov.cn/col/col38595/index.html, accessed on 15 February 2022); Shanghai Statistics Bureau (http://tjj.sh.gov.cn/tjnj/index.html, accessed on 15 February 2022); Sichuan Statistics Bureau (http://tjj.sc.gov.cn/scstjj/c105855/nj.shtml, accessed on 15 February 2022); Yunnan Statistics Bureau (http://stats.yn.gov.cn/tjsj/tjnj/, accessed on 15 February 2022); Zhejiang Statistics Bureau (http://tjj.zj.gov.cn/col/col1525563/index.html, accessed on 15 February 2022); Chongqing Statistics Bureau (http://tjj.cq.gov.cn/zwgk_233/tjnj/, accessed on 15 February 2022); and China’s economic and social big data research platform (https://data.cnki.net/, accessed on 15 February 2022).

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
