# Peer review of "Impact of Industrial Structure Upgrading on Green Total Factor Productivity in the Yangtze River Economic Belt"

_ijerph, 2022, doi:10.3390/ijerph19063718_

Round 1
Reviewer 1 Report
Please see attached PDF file.

Reviewer 2 Report
The subject of the article is very interesting and topical. It is adequate and relevant to the journal theme and scope, including the Ecology and the Environment section, but also the broadly understood ecology and engineering topics. It addresses critical issues related to environmental quality and indirectly public health.
The development of industry is usually associated with an increase in its environmental impact. The authors analyze an important aspect, which is the analysis of the development of an important and rapidly developing industrial region of the country (Yangtze River economic belt), where negative effects can bring severe, negative, short-term and long-term environmental effects. The search for opportunities to improve the effects of human industrial activity will always be an important research area. The analysis of such factors, especially in such a region, is extremely valuable.
I hope this paper will be published, however, I suggest considering following improvements:
- Introduction is too weakly supported by references.
- The introduction lacks a clear definition of the research gap and the research questions (hypothesis) to which researchers are looking for answers (they want to prove), goals. That would confirm the correctness of the further structure of the Review of literature section. According to the formal requirements of the Journal, the main conclusions should be here. It is also a good practice to include in this section the description of the further structure of the content of the article. It can be briefly announce what has been done in individual chapters, describing how the article is organized.
- The article lacks a coherent description of the stages of the research. These elements partially appear in different places in the content and it may be difficult for the reader to understand them properly.
- The article does not discuss the limitations of the conducted research. There is no information on the limitations of the obtained test results.
- There is no discussion in the Research Conclusions and Suggestions section.
- Some of the content of the article should be supplemented with references: e.g., 25-41 (including figures 1-2), lines 56-64, Section Research Conclusions and Suggestions, section Introduction.
- Page 5 - no footnote numbering.
- Errors in the form of references. As required, references should be numbered in order of appearance and indicated by a numeral or numerals in square brackets, before the punctuation —e.g., [1] or [2,3], or [4-6]. There are errors in the content, e.g., in lines: 54, 124, 158, 206, footnote on page 5, 43-35.
- Figures 2,3 needs a legend, and units on the chart axes.
- The list of references should be carefully reviewed, and minor errors corrected.
- Double numbering of formulas reduces the readability of the content.
- Line: 277 - there is no indication to which figure the text in parentheses corresponds.
I would like to thank the authors for their work in developing the article and see this manuscript published but considering at least the above issues.
Author Response
Response to Reviewer 2’s Comments
- Introduction is too weakly supported by references.
Response 1: Introduction has been revised and the relative references has been marked in the paper from line 24 to line 82.
- The introduction lacks a clear definition of the research gap and the research questions (hypothesis) to which researchers are looking for answers (they want to prove), goals. That would confirm the correctness of the further structure of the Review of literature section. According to the formal requirements of the Journal, the main conclusions should be here. It is also a good practice to include in this section the description of the further structure of the content of the article. It can be briefly announce what has been done in individual chapters, describing how the article is organized.
Response 2: These contents has been described in the new version from line 43 to line 82.
- The article lacks a coherent description of the stages of the research. These elements partially appear in different places in the content and it may be difficult for the reader to understand them properly.
Response 3: A coherent description of the stages of the research is revised in the paper as below: This paper aims to study the impact of industrial structure upgrading of the YREB on green total factor productivity (GTFP). We collected the relevant data of 11 provinces (municipalities) in the YREB from 2005 to 2019 for analysis. Firstly, considering the undesirable output such as environmental pollution, we use the super efficiency SBM model to calculate the value of GTFP in 11 provinces (municipalities). We find that GTFP presents regional characteristics in the YREB. Then, we establish a model to study the impact of the upgrading and rationalization of industrial structure on GTFP. The results show that the industrial structure upgrading has a significant positive impact on GTFP, while the industrial structure rationalization of industrial structure has no significant impact on GTFP. At the same time, environmental regulation, as a regulatory variable, also has a significant impact on GTFP. After further analysis, it is found that the impact of industrial structure upgrading and environmental regulation on GTFP in the eastern region is significantly higher than that in the western region. At the same time, we also observed that urbanization also promoted the improvement of GTFP. Finally, we review and prospect the analysis. The YREB should promote the upgrading of industrial structures, protect the environment and promote urbanization to improve GTFP.
This paper makes contributions to the literature from the following aspects. Firstly, our paper provides new evidence for understanding the relationship between the upgrading of industrial structure and GTFP. Promoting the upgrading of industrial structure upgrading can promote energy conservation and environmental protection, so as to improve GTFP. Secondly, this paper provides suggestions for the development of the YREB. We believe that the economic development of provinces (municipalities) in the YREB is unbalanced, and the industrial structure has different effects on GTFP in different regions. Therefore, provinces and municipalities should adjust measures to local conditions to promote industrial structure upgrading and environmental protection.[4-5]
- The article does not discuss the limitations of the conducted research. There is no information on the limitations of the obtained test results.
Response 4: The limitations and future research are described in the paper as below: Although this paper provides sufficient evidence for the impact of industrial structure upgrading of the YREB on GTFP, it also has limitations for further research. Firstly, this study uses the data envelopment analysis method to measure the GTFP. The result is a static efficiency value, so the result is not comprehensive enough, and it will be improved in future research. Secondly, this study divides industrial structure upgrading into industrial structure upgrading and industrial structure rationalization. In further research, the calculation of industrial structure upgrading can be further expanded. Finally, this paper takes 11 provinces (municipalities) in the YREB as the research object and draws some conclusions. Among these 11 provinces (municipalities), there is a large gap among the constituent cities. Further exploration and research should be made on the adjustment of industrial structure.
- There is no discussion in the Research Conclusions and Suggestions section.
Response 5: In the Research Conclusions section we discussed that GTFP values of 11 provinces (municipalities) appear different feature, the industrial structure upgrading index of 11 provinces (municipalities) show a changing trend, and the impact of industrial structure upgrading and environment regulation on the GTFP. In the Countermeasures and suggestions section according the empirical analysis we suggest that promote the industrial structure upgrade, promote environmental protection, and push forward urbanization of YREB.
- Some of the content of the article should be supplemented with references: e.g., 25-41 (including figures 1-2), lines 56-64, Section Research Conclusions and Suggestions, section Introduction.
- Page 5 - no footnote numbering.
- Errors in the form of references. As required, references should be numbered in order of appearance and indicated by a numeral or numerals in square brackets, before the punctuation —e.g., [1] or [2,3], or [4-6]. There are errors in the content, e.g., in lines: 54, 124, 158, 206, footnote on page 5, 43-35.
- Figures 2,3 needs a legend, and units on the chart axes.
- The list of references should be carefully reviewed, and minor errors corrected.
- Double numbering of formulas reduces the readability of the content.
- Line: 277 - there is no indication to which figure the text in parentheses corresponds.
Response 6-12: Great appreciate for your careful and kind recommend. The above-mentioned places in the paper have been revised.

Round 2
Reviewer 2 Report
I accept the changes made by the authors. The final layout of the content, the form of presentation, description and inference reflect the formal requirements, but also the individual approach of the authors and they may differ slightly from the opinions of others. In my opinion, the discussion of the results is not sufficiently related to the literature review. However, the article meets the requirements and can be published in its current version. I thank the authors for their work in developing the article and I wish them good luck.